# Seafood Waste Management Status in Bangladesh and Potential for Silage Production

**Md Jakiul Islam** [1,2,*] and **Omar Riego Peñarubia** [3]

1   Faculty of Biology and Chemistry, University of Bremen, 28359 Bremen, Germany
2   Faculty of Fisheries, Sylhet Agricultural University, Sylhet 3100, Bangladesh
3   Fisheries Division, Food and Agriculture Organization of the United Nations (FAO), 00153 Rome, Italy; OmarRiego.Penarubia@fao.org
*   Correspondence: mdjislam@uni-bremen.de or jakiulmi@sau.ac.bd

**Abstract:** Frozen shrimp and fish are the second most valuable export items from Bangladesh. Thus, in processing industries, a considerable amount of seafood waste is produced every year. Neglecting seafood waste leads to serious forms of wastage. The purpose of this survey-based study was to estimate the amount of seafood waste produced and understand the existing waste management practices in Bangladesh. Potential for seafood waste-based silage production and its utilization were also studied. Across the seafood industry, around 43,321 tons of seafood waste are produced every year. The highest amount of seafood waste is produced in Khulna, followed by Chittagong, Cox's Bazar, Dhaka, and Sylhet. Local people consume a portion of fresh shrimp carapace and heads and gills of large fish. A portion of seafood waste is also used to feed aquaculture species. Moreover, parts of dried shrimp shells, appendages, and fish scales, air bladders, and fins are exported to some Asian countries. The prospect of fish silage production constitutes a promising new development for animal feed production in Bangladesh. The availability of waste materials from seafood processors and the demand from feed millers favor the conditions for silage production. However, in order for the seafood waste-based silage industry to flourish, the establishment of supply chains for seafood waste and end products (silage) is required. Studies on growth performance, muscle quality, and digestibility of animal feed with silage-based diets are required for farmed species.

**Keywords:** fish waste; utilization; management; silage; Bangladesh

## 1. Introduction

In recent decades, aquaculture has been the world's fastest-growing food-production sector, providing a protein-rich supplement [1]. Bangladesh has been among the top five fish-producing countries [2,3], and production has increased by 53 percent since 2009 [1]. By 2022, this country will be among the top four fish-producing countries [4]. The country has a coastal area of 2.30 million hectares and a 714 km coastline along the Bay of Bengal and supports large numbers of artisanal and coastal fisheries. Moreover, post maritime boundary settlement with Myanmar and India, marine capture from the Bay of Bengal is likely to be increased manifold [5,6]. Seafood is the second most valuable export commodity from Bangladesh. It contributes almost 3.65% to gross domestic product (GDP), 25.30% of gross agriculture products, and 2.0% of total export earnings, worth $526.45 million USD [3,7–10]. This contribution would have been higher if the seafood waste had also been used effectively. Among the different types of seafood, frozen shrimp and fish are the main seafood export items [7–10]. Public and private organizations have installed more than 100 shore-based export-oriented fish processing plants at Chittagong, Khulna, Dhaka, and Sylhet divisions. Out of 100 processing plants, 76 are European Union (EU) compliant, and 30 plants are USFDA green ticketed [11] with a total of 350,000 tons processing capacity [8]. Bangladesh's seafood processing industries have the vast potential for both vertical and horizontal expansion [6,7,10,12].

Seafood industries produce a considerable quantity of processing byproducts, which are mainly inedible and constitute approximately 40–60% of the wet weight [13–15]. Seafood waste is mainly composed of heads, viscera, bones, and scales and is rich in lipids, proteins, and other bioactive compounds [16]. The ever-increasing production of these byproducts without utilization has resulted in environmental pollution [17]. Waste management activities include collection, transportation, processing, utilization, and disposal [18]. Monitoring and controlling of waste management methods are also considered important [19]. Inappropriate waste management (e.g., open dumping, indiscriminate littering) causes environmental pollution and forms breeding grounds for insects and vermin, posing significant public health risks [20]. Consequently, waste management practice is coming under strict regulations due to environmental issues and has become an increased cost burden for the seafood industries [21,22]. The proper management and utilization of seafood processing waste and its conversion into value-added products will lead to better resource utilization and profit maximization [23,24]. This also results in significant environmental and economic improvement. In contrast, underutilization of byproducts leads to loss of potential revenues and additional disposal costs [20,23].

Seafood waste has many applications, among which the most important are animal feed, biodiesel, and biogas [14,25,26]. Furthermore, seafood waste contains several bioactive compounds like chitin, collagens, biofilms, pigments, amino acids, and fatty acids [27–29]. Efficient utilization of seafood waste can only be achieved when properly utilized and no wastage is allowed [13,30–32]. Value addition, product diversification, utilization, and efficient management of seafood waste will generate more profits [33–35]. For efficient seafood waste management, information on the amount, types, and existing management status is essential. However, for many countries including Bangladesh, information on the amount of seafood waste produced and the present management status is still lacking [36]. Due to biological instability, high moisture content, high enzymatic activities, and rapid auto-oxidation, utilization of seafood waste is difficult. Thus, waste management and disposal in the seafood processing industry pose problems in terms of environmental protection and sustainability [31,37]. In less technology-intensive countries, seafood waste is usually used to produce animal feed ingredients such as fishmeal and silage [23,38–40]. The seafood waste-based silage production process is relatively simple and is much less expensive than producing fishmeal [41–43]. Moreover, seafood waste (fish and shrimp)-based silage production fits best at places where the supply chain is not well organized, and the amount and availability of waste are not sufficient to justify the operation of a fishmeal plant [41].

Fish silage is a protein-rich liquid produced from enzymatic hydrolysis of fish byproducts and bycatch [32,43,44] that is rich in a mixture of hydrolyzed proteins, lipids, and minerals. It is easily digestible and absorbed by terrestrial and aquatic animals [35,45,46]. Silage is prepared by combining ground processing waste material with inorganic acids to attain a lower pH (<4.0) where bacterial growth is inhibited, and spoilage is prevented. Thus, silage can be stored for years and used when needed [47]. Details of the fish and shrimp silage production techniques are available in the literature [48–50]. Fish and shrimp silage has the high potential for use in aquaculture due to the similarity in raw materials, protein source, and low operation cost compared to fishmeal [45,51–53]. Fish and shrimp silage can act not only as a useful feed ingredient but also as a feed additive [32]. Fish silage-based diets fed to *Litopenaeus vannamei* [51,54,55]; Nile tilapia, *Oreochromis niloticus* [22,56–58]; pacu, *Piaractus mesopotamicus* [46,59]; broiler chicken [39,40,60]; quails, *Coturnix japonica* [61]; and rohu, *Labeo rohita* [53] animals showed improved growth, digestibility, and physiological fitness. Therefore, the application of silage-based diets for aquaculture and livestock production is promising. However, for Bangladesh, there are insufficient data available on the amount of seafood waste generated and the present status of waste management and utilization. Information on the potential for seafood waste-based silage production is lacking. Thus, a better understanding of the amount of seafood waste produced and its management status is essential to prepare an effective

seafood waste management plan for silage production. In this study, quantitative and qualitative estimation was conducted to identify the amount of seafood waste generated and to understand existing management practices. Furthermore, animal feed producers were also surveyed to assess the potential for fish silage production and utilization.

## 2. Materials and Methods

### 2.1. Selection of Study Area

The study was conducted at four major seafood processing areas, Khulna, Chittagong, Dhaka, and Sylhet (Figure 1), using semi-structured questionnaires (Supplementary Files 1 and 2) from November 2017 to March 2018. Out of 102 seafood industries, a total of 55 (EU and USFDA certified) were surveyed (Table 1). Chittagong and Khulna are the major fishing and fish landing centers located in the Southeast and Southwest of Bangladesh, respectively, where most of the seafood processing industries are located. Thus, more emphasis was placed on these two areas when conducting the survey (Table 1). Moreover, 12 fish feed mills and 160 aquaculture farmers were also surveyed to explore the feasibility of seafood silage production and the utilization as a potential protein source for fish and shrimp feeds.

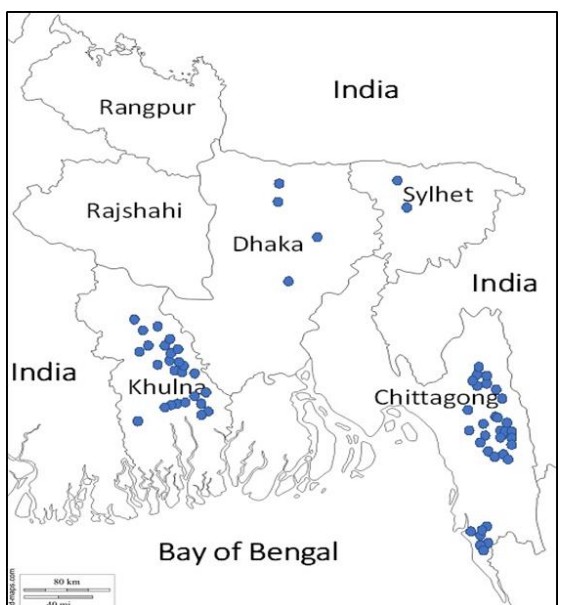

**Figure 1.** Locations of surveyed seafood industries in Khulna, Chittagong, Dhaka, and Sylhet.

**Table 1.** Locations of seafood processing industries in Bangladesh and distribution of surveyed industries.

| Location | * Number of Processing Industries | No. of Industries Surveyed | No. of Fish Feed Industries Surveyed | No. of Fish Farmers Surveyed |
|---|---|---|---|---|
| Khulna Division | 58 | 26 | 7 | 70 |
| Chittagong Division | 38 | 25 | - | 55 |
| Dhaka Division | 4 | 2 | 7 | 25 |
| Sylhet Division | 2 | 2 | - | 10 |
| Total | 102 | 55 | 12 | 160 |

* Data source: Bangladesh Frozen Food Exporter Association (BFFEA, 2017).

### 2.2. Field Investigations and Survey

For this study, in-person and key informant interviews were employed with a semi-structured questionnaire survey method to collect information on waste estimation, present management status, and silage production potential. Secondary information from various sources was also collected and reviewed to cross-check and validate the present findings.

The field surveys and field investigations also enabled visual assessment of seafood waste management practices and needs. Before starting the interview, each respondent was briefed about the study objectives.

### 2.3. Statistical Analysis and Tools Used

Descriptive statistics and chi-squared ($\chi^2$) tests were used to summarize and analyze the survey data, respectively. Factor analysis (FA) was conducted to elucidate stakeholders' perception and willingness to properly utilize seafood waste and silage. The FA tool interprets the perception to include experience based on subjective perception, experience, and information available to the stakeholders on products [62,63]. R software (Version 3.2.4) was used for data analysis and map creation.

## 3. Results

### 3.1. Volume and Value of Seafood Waste Generated

Across the seafood industry in Bangladesh, around 43,320.88 tons per year of seafood waste is produced, worth $13.73 to $44.09 million USD. Shrimp and fish waste are being produced at the highest rate at 23,190.24 and 17,605.71 tons per year, respectively. Both fish and shrimp waste are produced most in Khulna, followed by Chittagong, Cox's Bazar, Dhaka, and Sylhet (Table 2). Fish markets in Khulna and Chittagong have an annual waste throughput of over 2500 tons, much of which is generated from retailers' purchases.

**Table 2.** Estimated amount and values of shrimp and fish waste (tons per year) generated by the seafood processing industries in Bangladesh [1].

| Waste Type | Khulna | Chittagong | Cox's Bazar | Dhaka | Sylhet | Total | Value (Million USD) |
|---|---|---|---|---|---|---|---|
| Industrial shrimp waste | 12,015.69 | 7965 | 2154.6 | 771.12 | 283.77 | 23,190.24 | 8.70 to 28.99 |
| Industrial fish waste | 7965 | 5310 | 923.4 | 1799.28 | 1608.03 | 17,605.71 | 4.40 to 13.20 |
| Other wholesalers/processors | 600 | 650 | 100 | 850 | 325 | 2525 | 0.63 to 1.89 |
| Subtotal | 20,580.69 | 13,925 | 3178 | 3420.4 | 2216.8 | 43,320.88 | 13.73 to 44.09 |

[1] Fish waste was calculated using a ratio of 20% to 40% of raw fish. In the case of shrimp waste calculation, the ratio was 35% to 50% of raw shrimp.

### 3.2. Product and Species-Specific Amount of Seafood Waste Generated during Processing

Seafood undergoes numerous processing steps such as filleting, heading, gutting, skinning, and cutting. Around 50% to 60% of waste is generated from the shrimp processing chain, whereas the value is 20% to 45% for finfish (Table 3). Head, skin, viscera, carcass, eggs, roe, and trimmings contribute to generating waste (Table 4).

**Table 3.** Seafood waste generated from different species and methods used.

| Processing Method | * Raw Fish (g) | * Waste Type | * Solid Waste (g) |
|---|---|---|---|
| Frozen *Penaeus* spp. | 1000 | Head and appendages | 350 to 450 |
| | | Shell | 50 to 100 |
| Frozen *Macrobrachium* spp. | 1000 | Head and appendages | 450 to 550 |
| | | Shell | 60 to 70 |
| | | Skin and scale | 40 to 60 |
| Fish filleting | 1000 | Heads | 150 to 260 |
| | | Bones, carcass | 160 to 280 |
| | | Gut, viscera | 100 to 150 |
| Dried fish | 1000 | Scale | 30 to 55 |
| | | Entrails | 80 to 120 |
| Salted and fermented fish | 1000 | Scale | 30 to 55 |
| | | Entrails | 85 to 120 |
| Salted dehydrated fish | 1000 | Scale | 30 to 55 |
| | | Entrails | 85 to 120 |

**Table 3.** *Cont*.

| Processing Method | * Raw Fish (g) | * Waste Type | * Solid Waste (g) |
|---|---|---|---|
| Dried brackish and seawater shrimp | 1000 | Head | 120 to 150 |
| | | Shell | 30 to 60 |
| Dried freshwater shrimp | 1000 | Head | 100 to 130 |
| | | Shell | 10 to 35 |
| Dried and smoked shrimp/prawn | 1000 | Head, shell, and appendages | 100 to 180 |
| *Sepia* spp. | 1000 | Tail, appendages, skin, ink, and blood | 450 to 550 |
| Sharks, skates, and rays | 1000 | Entrails, trimming leftovers | 300 |
| Scaling of white fish | 1000 | Scales | 20 to 70 |
| De-heading of white fish | 1000 | Head and debris | 270 to 320 |
| De-headed fish filleting | 1000 | Frames and offcuts | 190 to 310 |
| Filleting of un-gutted fish | 1000 | Entrails, tails, frames, and heads | 380 to 420 |
| Skinning of white fish | 1000 | Skin | 40 to 60 |

* Source: present study.

**Table 4.** Average proportion (in general) of fish byproducts.

| Portion | Percentage Found in the Present Study, % |
|---|---|
| Head | 9 to 20 |
| Skin | 1 to 4 |
| Viscera | 12 to 18 |
| Trimming | 8 to 17 |
| Carcass | 9 to 15 |
| Eggs, roe | 2 to 10 |

### 3.3. Present Status of Seafood Waste Management

Fresh shrimp and prawn heads with meat are collected by third parties for local consumption and to feed farm fish. For fish wastes, heads (and sometimes gills) of large fish are sold for local consumption by local people. Shrimp waste and fish entrail wastes are sold to local fish farmers to feed farm fish (Figure 2). Air bladders, scales, fins, shrimp heads, and appendages are dried, packed, and exported to China, Thailand, and Vietnam. Moreover, shark fins, viscera, and swim bladders of large marine and freshwater fish are exported to some Asian countries. (See Table 5) However, the adoption of other available processing and utilization methods is also possible (Figure 2 and Table 6).

**Table 5.** Price of different types of fish and shrimp wastes.

| Waste Type | Price (* BDT per Kg) |
|---|---|
| Shrimp head | 25 to 140 |
| Shrimp shell | 20 to 80 |
| Fish head (large) | 60 to 120 |
| Fish head (small) | 20 to 60 |
| Fish scale, gill, and entrails | 15 to 35 |

* BDT = Bangladesh currency (taka); 1 USD = 84.93 BDT (www.xe.com/currencyconverter/, accessed on 12 August 2020).

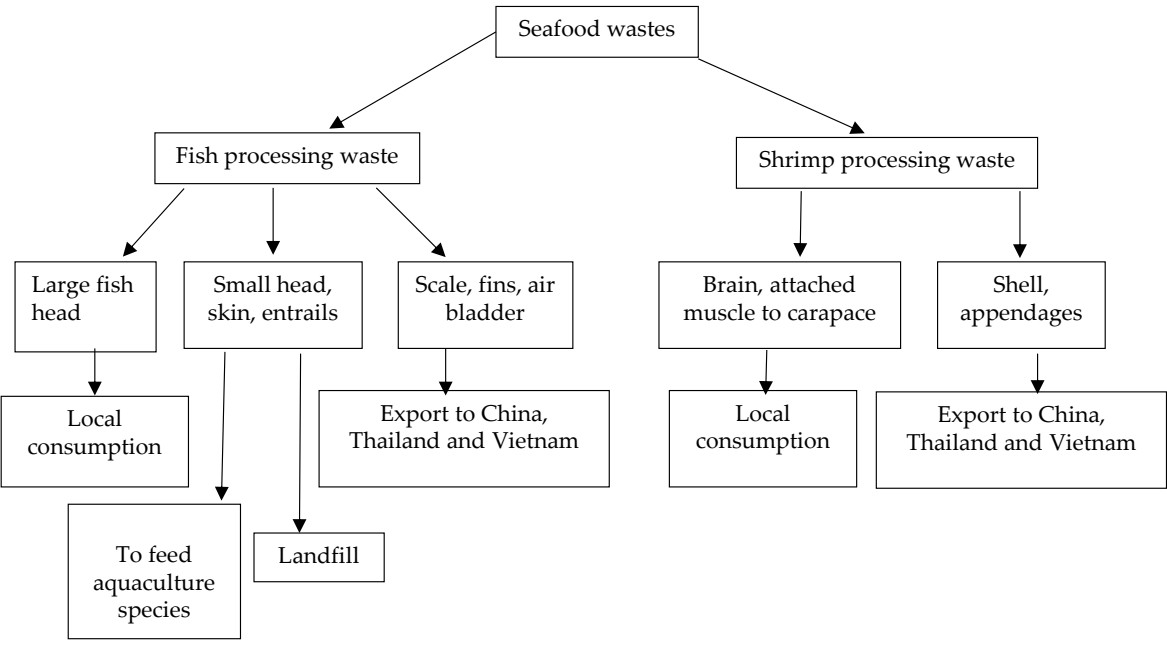

**Figure 2.** Existing seafood waste management practices in Bangladesh.

**Table 6.** Present usage, possible uses of seafood waste.

| * Processing Method | * Waste Type | * Current Usage | Possible Usage of Waste | References |
|---|---|---|---|---|
| *Penaeus* spp./*Macrobrachium* spp. | Head and appendages | Head: human consumption, direct use as fish feed, dried head. Appendage: exported, fertilizer, sometimes landfill | Protein-rich portion as an animal feed supplement, residual solid as a source of calcium carbonate, and production of chitin and chitosan. Head of large individuals for human consumption | [23,31,64–66] |
| | Shell | Exported to Thailand, China, Vietnam | Production of chitin and chitosan | |
| Finfish filleting | Skin/scale | Exported to Thailand, China, Vietnam | Feedstuff in fish and animal diets as a protein substitute, e.g., fish silage, fishmeal, fish hydrolysates. Head for consumption in some parts of the world (e.g., Asia). Fish leather and gelatin, biodiesel | [16,26,67–72] |
| | Heads | Direct consumption by local people | | |
| | Bones/carcass | Fishmeal, landfill | | |
| | Gut/viscera | Directly used to feed farmed fish, landfill | | |
| Dried fish | Scales | Exported to Thailand, China, Vietnam | Deproteinized fish scales, collagen, hydroxyapatite, chitin, and chitosan, fertilizer | [26,67,73–75] |
| | Entrails | Directly used to feed farmed fish, landfill | Fishmeal, fish silage | |
| Salted and fermented fish | Scale | Exported to Thailand, China, Vietnam | Heavy metal and pigment removal from wastewater, protein recovery, chitin, and chitosan | [76–78] |
| | Entrails | Directly used to feed fish, landfill | Fishmeal, fish silage, enzymes | |
| Salted dehydrated fish | Scale | Exported | Ornaments, collagen, hydroxyapatite, chitin, and chitosan | [78–80] |
| | Entrails | Used to feed farmed fish, landfill | Fishmeal, fish silage, enzymes | |
| Dried shrimp | Head and appendages | Fishmeal | Carotenoids, chitin, and chitosan | [77,81] |
| Dried and smoked shrimp/prawn | Head, shell, and appendages | Fishmeal | Fishmeal, animal feed ingredients | |

**Table 6.** *Cont*.

| * Processing Method | * Waste Type | * Current Usage | Possible Usage of Waste | References |
|---|---|---|---|---|
| *Sepia* spp. | Tail, appendages, skin, ink, and blood | Directly used to feed farmed fish | Silage, fishmeal | [67,82] |
| Sharks, skates, air bladders, and rays | Entrails, trimming leftovers | Fishmeal directly used to feed fish | Fishmeal, soup powders, fish oil, liver oil | [64,82] |
| Scaling of white fish | Scale | Exported, landfill | Ornaments, collagen, hydroxyapatite, chitin, and chitosan | |
| De-heading of white fish | Head and debris | Head for human consumption, debris is used to feed fish, landfill | Head (consumption, fishmeal), fishmeal, fertilizers | [31,64,71,76,77,80,83] |
| De-headed fish filleting | Frames and offcuts | Fishmeal, feeding farmed fish | Value-added fish products, fishmeal | |
| Filleting of un-gutted fish | Entrails, tails, frames, and heads | To feed farmed fish, head for local consumption | Fish oil, fish silage, fishmeal, direct consumption, direct feeding to fish | |
| Skinning of white fish | Skin | To feed farmed fish, landfill | Leather and gelatin | |

\* Present study.

### 3.4. Utilization Potential of Seafood Waste in Bangladesh

3.4.1. Perception of Fish and Shrimp Silage Technology in Bangladesh

More than 83% of the surveyed seafood processing and animal feed companies were not aware of the production and utilization of fish and shrimp silage. However, around 16% of seafood processing industries had an idea of fish and shrimp silage, though, for animal feed industries, this number was only 7.14% (Table 7).

**Table 7.** Awareness of fish and shrimp silage by seafood processors and animal feed producers.

| Type of Industry | Awareness of Fish and Shrimp Silage | | |
|---|---|---|---|
| | **Yes** | **No** | **Unsure** |
| Seafood processing industries | [a] 16.36% | [b] 83.64% | [c] 5.45% |
| Animal feed industries | [a] 7.14% | [b] 85.71% | [c] 7.14% |

Superscripts of a, b, c at each row indicate significant difference, $p < 0.05$.

3.4.2. Potential for Fish Silage Production and Its Utilization

Most seafood processors and animal feed producers have shown their interest in fish and shrimp silage production and utilization, respectively. However, substantial and skillful hands-on training is required for them (Table 8). Seafood processors, feed producers, fish farmers, and local people showed their strong interest in fish silage (Table 9). Seafood waste-based silage can be produced within the industry premises or nearby. Fish and silage can be prepared either separately or combined and then transported to animal feed millers and local farmers who prepare farm feed based on their needs. Feed industry-made feeds can be distributed through the existing aquaculture feed supply channel. Based on the findings of the current study, a supply chain suitable for fish silage production and distribution is proposed (Figure 3).

**Table 8.** Willingness of seafood processors and animal feed industries to produce and use fish and shrimp silage.

| Opinion on Fish Silage Use | Seafood Processing Industries | Fish and Animal Feed Industries |
|---|---|---|
| Willing to produced fish and shrimp head silage | 94% | 90% |
| Willing to use fish and shrimp head silage as the protein source | - | 96% |
| Training required for the production and use of silage | 96% | 98% |

**Table 9.** Stakeholder's preferences for fish waste production and usage in Bangladesh.

| Stakeholders | Usage Preferences | | | | | | |
|---|---|---|---|---|---|---|---|
| | Fish Silage | Fishmeal | Edible Part Consumption | Fertilizer | Fish Oil | Pharmaceutical Items | Landfill |
| Seafood industries/processors | <u>0.824</u> | 0.752 | −0.148 | 0.256 | −0.124 | 0.345 | −0.21 |
| Aquaculture feed industries | <u>0.925</u> | 0.793 | −0.256 | 0.187 | 0.174 | −0.265 | −0.369 |
| Fish farmers | <u>0.852</u> | 0.831 | 0.425 | 0.281 | −0.459 | −0.695 | 0.321 |
| Shrimp farmers | <u>0.822</u> | 0.791 | 0.425 | 0.281 | −0.459 | −0.695 | 0.321 |
| Byproduct Consumers | 0.563 | 0.441 | <u>0.891</u> | 0.249 | 0.256 | −0.264 | −0.214 |

Note: Parameters with the highest factor loadings are underlined.

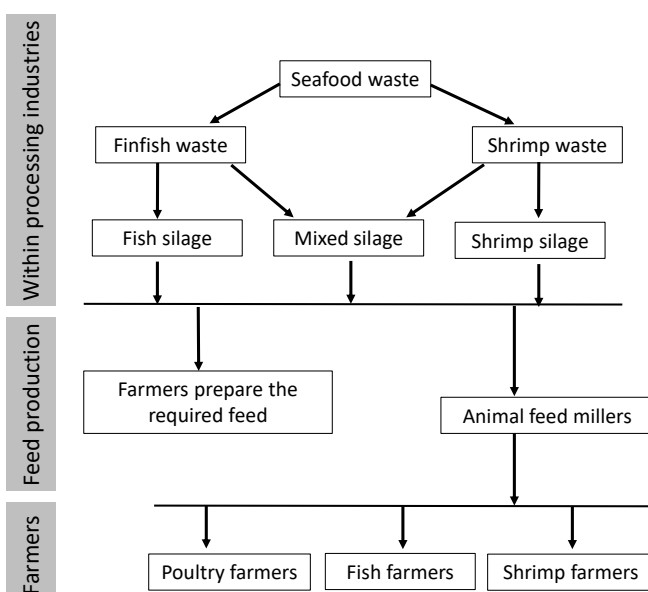

**Figure 3.** Proposed supply chain for fish silage production and utilization in Bangladesh.

*3.5. Environmental Impacts of Existing Fish and Shrimp Waste Management*

Seafood waste creates visual environmental pollution followed by air pollution. Local people complained about insects, rodents, and water pollution. Some of the problems created include water pollution, fouled beaches, insect/rodent infestations, and obnoxious odors. All respondents agreed that seafood waste contributed to polluting the environment (Table 10). Waste produced by seafood processors ends up in the surrounding environment and pollutes waterways, rivers, canals, fresh groundwater supply, indigenous flora, and fauna. Seafood waste significantly contributes to industry-generated organic waste. Environmental concerns and legal limitations on its disposal are now being recognized as problems. Fish and shrimp offal generated by seafood processing plants has occasionally

caused health and environmental concerns owing to improper utilization and poor storage, handling, and disposal practices. In terms of sustainability, the present seafood waste management systems are not environmentally friendly or economically beneficial.

**Table 10.** Factor analysis of environmental and social problems that arise from the fish industry waste.

| Stakeholders | Visual Environmental Pollution | Air Pollution (Bad Odor) | Insects/Rodents | Water Pollution (Eutrophication) |
| --- | --- | --- | --- | --- |
| Seafood industries/processors | <u>0.918</u> | 0.752 | 0.564 | 0.759 |
| Aquaculture feed industries | 0.856 | <u>0.893</u> | 0.669 | 0.687 |
| Fish farmers | <u>0.882</u> | 0.831 | 0.587 | 0.624 |
| Shrimp farmers | <u>0.932</u> | 0.891 | 0.574 | 0.714 |
| Byproduct/edible waste consumers | <u>0.973</u> | 0.741 | 0.647 | 0.632 |

Note: Parameters with the highest factor loadings are underlined.

## 4. Discussion

Shrimp and fish are the second-largest export items in Bangladesh after readymade garments [5]. Every year a considerable quantity of processing byproducts are produced [83] in the seafood processing industries [84,85]. However, information on the amount of seafood waste generated in Bangladesh was not available, which made it difficult to compare the findings of this present study. To the best of our knowledge, this is the first baseline study on the amount of fish waste produced and present waste management status in Bangladesh. In Chittagong and Khulna regions, the highest amount of shrimp and fish waste is produced. Seafood industries processed both captured (freshwater, marine water) and cultured shrimp and finfish [8]. Fish landed at Chittagong and Khulna through 76 and 12 landing centers, respectively [86], from farms, wild stocks, and marine catch. These fish are exported as gutted, beheaded, and scaled products, which generates a substantial amount of waste. The current survey indicated that significant quantities and varieties of finfish and shellfish wastes are being produced. An interesting corollary of increased demand for exports of shrimp shell and fish scale to other Asian countries has been contributing slightly to a reduced volume of wastes. The survey also identified opportunities where a significant amount of waste can be used to produce sustainable aquaculture and livestock feeds.

An essential aspect of the present study was the quantitative evaluation of seafood waste for further processing. Such evaluation was difficult due to poor recordkeeping. Thus, to cross-check and validate the present study findings, the estimated seafood amount was compared with seafood export data. In 2017, the total exported amount was 68,935 tons [11], of which 65% was shrimp, 27% was fish, and the remaining 8% was fins and air bladders [8]. Another approach was followed to quantify the volume and value of fish and shrimp waste from the export data of the Bangladesh Frozen Food Export Association [8]. The estimated volume and value of the present findings agree with the 2016-2017 export data (Table 11). Fresh shrimp and fish heads were found to be consumed by local people and used to feed farm fish. However, the fish feeding process was found wasteful and inefficient (Supplementary File 3). Direct application of seafood waste as fish feed acts as a threat to sustainable aquaculture. The used raw seafood waste was produced in different environments, thus increasing the chance to introduce potentially harmful pathogens (algae, amoebae-protozoans, and viruses) to which farmed fish and shrimp can be susceptible [31]. Excessive bio-deposition of applied waste as feed puts aquaculture at risk by deteriorating water quality [9,87]. A portion of the fish air bladders, scales, fins, and shrimp heads and appendages (both spoiled and fresh) are dried and exported to China, Thailand, and Vietnam. However, sometimes, specifically during abundant supply, fish entrails, scales, gills are carried away and discarded as landfill. However, the estimation of the waste export volume was not possible for lack of recordkeeping. Seafood wastes are an important source of environmental pollution [37]. Environmental impacts from seafood waste appear at all stages of the product lifecycle, from the collection of resources, processing, and

disposal [23]. Fish waste affects not only the surrounding area but is also directly affected by the effluent discharge. However, it can also alter wider areas at different ecosystem levels and the biomass, density, and diversity of the benthos, plankton, and nekton in the receiving water body [34].

**Table 11.** Fish and shrimp export from Bangladesh (2001 to 2017), estimated waste, and value.

| * Year | * Item | * Quantity (Tons) | * Value (Million USD) | [1] Estimated Waste Volume (Tons) | [2] Price (Million USD) |
|---|---|---|---|---|---|
| 2001–2002 | Shrimp and Fish | 40079.38 | 276.11 | 10019.85 to 24047.63 | 3.13 to 24.05 |
| 2002–2003 | Shrimp and Fish | 33370.76 | 321.81 | 8342.69 to 20022.46 | 2.61 to 20.02 |
| 2003–2004 | Shrimp and Fish | 38319.45 | 390.25 | 9579.86 to 22991.67 | 2.99 to 22.99 |
| 2004–2005 | Shrimp and Fish | 43594.72 | 420.74 | 10898.68 to 26156.84 | 3.41 to 26.16 |
| 2005–2006 | Shrimp and Fish | 48924.43 | 459.11 | 12231.11 to 29354.66 | 3.82 to 29.35 |
| 2006–2007 | Shrimp and Fish | 50870.34 | 515.32 | 12717.59 to 30522.21 | 3.97 to 30.52 |
| 2007–2008 | Shrimp and Fish | 50507.46 | 534.07 | 12626.87 to 30304.48 | 3.95 to 30.30 |
| 2008–2009 | Shrimp and Fish | 53210.87 | 454.53 | 13302.72 to 31926.53 | 4.16 to 31.93 |
| 2009–2010 | Shrimp and Fish | 58880.77 | 437.4 | 14720.19 to 35328.47 | 4.60 to 35.33 |
| 2010–2011 | Shrimp and Fish | 81619.34 | 611.36 | 20404.84 to 48971.61 | 6.38 to 48.97 |
| 2011–2012 | Shrimp and Fish | 96265.83 | 579.72 | 24066.46 to 57759.50 | 7.52 to 57.76 |
| 2012–2013 | Shrimp and Fish | 92283.29 | 543.84 | 23070.82 to 55369.98 | 7.21 to 55.37 |
| 2013–2014 | Shrimp and Fish | 77165.07 | 638.19 | 19291.27to 46299.04 | 6.03 to 46.30 |
| 2014–2015 | Shrimp and Fish | 83347.53 | 568.03 | 20836.88 to50008.52 | 6.51 to 50.01 |
| 2015–2016 | Shrimp and Fish | 75178.33 | 535.77 | 18794.58 to 45107.00 | 5.87 to 45.11 |
| 2016–2017 | Shrimp and Fish | 68161.26 | 526.45 | 17040.32 to 40896.76 | 5.33 to40.90 |

* Data source: BFFEA (2017). [1] Fish waste was calculated using a ratio of 10% to 30% of raw fish. In the case of shrimp waste calculation, the ratio was 35% to 50% of raw shrimp. [2] Waste value was calculated based on Table 5.

Seafood waste is continuously gaining ground as a waste management field. Research has been carried out to convert these wastes into useful products [88–91]. Among the most prominent current uses of seafood waste are collagen and antioxidant isolation for cosmetics, biogas, biodiesel, fertilizers, chitosan, packaging materials (e.g., gelatin, chitosan), enzymes (e.g., proteases), feed ingredients (e.g., fish sauce, fishmeal, and fish silage) [30,31,92]. Most of the technologies known for the utilization of seafood waste are not economically attractive due to the high initial investment [92]. Among these options, some are technology-intensive and require higher investment costs. However, fish waste can be profitably used as fish feed through fishmeal and fish silage production [20], since seafood waste represents half of the raw material and could be an excellent source of low-cost nutrients [30]. In some countries, fishmeal is being traditionally produced from seafood waste, but this requires high investment, a high degree of coordination, a well-organized supply channel (with cold chain facilities), and an adequate volume of steady supply [16,23]. Seafood waste transportation to fishmeal plants is not always sustainable practice for the seafood processing industry [34,65,93]. In Bangladesh, these facilities are not available. Thus, it is essential to find suitable alternatives to manage waste within the industries or nearby, emphasizing energy savings, environmental concerns, and sustainability. Fish silage production can be a viable alternative, as it is an easy-to-make product that is less technology-intensive that requires low investment and little space [42,67,94]. Moreover, the advantages of silage production over fishmeal are: The process is virtually independent of the supply scale (e.g., amount, volume, and raw material quality); simple technology; little initial investment for large-scale production; reduced effluent and odor problems; and the product can be used locally [42,95]. In addition, fish and shrimp silage has good nutritional values and can therefore be used as an attractive alternative to fishmeal [46,57,96].

Seafood byproducts and waste-based silage production procedures are safe, cost-effective, and environmentally friendly [97]. The product has the excellent nutritional quality and can be sufficient for animal feeding [22,45,98,99]. To understand the potential for the production and utilization of fish and shrimp silage from seafood waste, we surveyed the feed producers and fish farmers. The majority of seafood processors were

unaware of fish silage and its potential as a profitable waste management method. The incorporation of seafood waste-based silage will be a new development in the production of animal feeds. Moreover, the use of fish silage in Bangladesh has not been studied yet. In the present study, most seafood industries and animal feed millers have shown their interest in fish and shrimp silage because both aquaculture and livestock farming are substantially dependent on high-cost feed inputs and exploring options for cheaper costs and improved feed conversion ratios [100–102]. However, for seafood processors, hands-on training is required to start fish and shrimp-based silage production. Seafood processors and feed millers are reluctant to produce and use silage for the following reasons: (i) a lack of fish and shrimp silage technology, (ii) reliability of fish waste supply for silage production and fish silage supply to the feed millers, and (iii) transport cost to carry fish waste and fish silage. Higher transport costs for fish silage than dry components (e.g., fishmeal) are cited as a disincentive [20,45]. While this proposition is not true on a per ton of protein basis, it discounts the fishmeal transport costs from abroad [103].

     Fish and shrimp silage is a liquid product produced from the whole fish and shrimp or parts, to which acids, enzymes, or lactic-acid-producing bacteria are added, with the liquefaction of the mass provoked by the action of enzymes from the fish [45,104]. Silage can be stored for years and used when needed [47]. The preparation of silage usually depends on locally available raw materials and conditions [105]. Organic acid, most preferably formic acid, is the best choice for silage preparation. Formic-acid-made silage is not excessively acidic and does not require neutralization before use and thus can be used directly [45,106]. The increasing demand and progressive scarcity of fishmeal in the international market boosted its price and launched attempts to reduce fishmeal in fish diets and the consequent search for alternative, acceptable, and digestible protein sources [106,107]. Fish and shrimp silage can be an excellent fishmeal alternative. This will be an incentive to utilize seafood and fish waste as a protein source for aquaculture and livestock [108] production in Bangladesh. Fish and shrimp silage is considered a valuable feed ingredient that has been shown to improve animal feed quality [32,44]. With fish and shrimp silage, 20–75% of fishmeal can be replaced during feed preparation [47]. The use of fish silage as a feed ingredient could make differences in some ways, e.g., by (i) reducing levels of waste: environmental impact; (ii) providing nutrients and bioactive components: animal health; (iii) economic gains: as an alternative to replacing expensive fishmeal [32,47]. Seafood waste-based silage production draws potential attention and has increased importance compared to fishmeal due to the simple and easy production technology and lower investment cost [52]. Feeding cost represents 50% of the operational cost in the aquaculture industry to ensure expensive dietary protein sources [109,110]. There is a shortage in the world production of fishmeal, which is the main protein source [46,100]. Currently, the increased demand for fishmeal for aquaculture is considered a possible deterrent for aquaculture growth [34]. Fish and shrimp silage can substitute for fishmeal without impacting animal growth. Research has shown fish silage as a good feedstuff for aquafeeds in terms of nutritional benefits and economic feasibility [30]. Inclusion of fish silage in feed increases the appetite and growth of farm animals [47]. For example, fish and shrimp silage-based diets used to feed Nile tilapia, *Oreochromis niloticus*, squilla, *Oratosquilla nepa*, and chicken had good growth performance equivalent to a fishmeal diet [45,98,108,111,112]. Likewise, fish silage-based diets supplied to Nile tilapia *O. niloticus* [113] and African catfish, *Clarias garipinus* provided 25% to 50% fishmeal replacement [97]. Digestibility and growth studies have shown that fish silage is highly digestible and can be used to replace up to 75% of fishmeal in aqua-feeds [52]. For shrimp, *Litopenaeus vannamei* fed with 25% to 50% silage-based diets showed higher growth performance [51,114] and improved gut health [54]. Fish silage not only acts as a feed ingredient but also an essential feed additive [32]. Moreover, 100% fishmeal replacement is also possible if fish silage is used along with other protein-rich plant ingredients, for example, soybean meal [51] and rice bran [113]. The free amino acids and peptides in the silage are pre-digested proteins, and the presence in the feed results in improved growth. Moreover, the organic acids in fish and shrimp silage have useful antimicrobial

properties, enabling livestock to perform better against diseases and in terms of mortality. Eventually, this could contribute to eliminating the use of non-therapeutic antibiotics in livestock feed [32,46,47].

In Bangladesh, there is a considerable gap between production origin and use of fish, shrimp, and poultry feed ingredients [70,115]. The incorporation of seafood byproduct-based silage could contribute to reducing the gap. Given the relatively wide geographical area (Khulna, Chittagong, Dhaka, Sylhet) covered by Bangladesh's seafood industry and the variability in the amount and composition of species involved, fish and shrimp silage can be one of the most suitable options to utilize seafood waste. With this goal in mind, a partnership is required between seafood and animal feed industries. In this study, it was acknowledged that if the utilization of seafood waste were to be successful on a broad scale, it would require a considerable level of coordination and cooperation among seafood companies and animal feed producers. Thus, a more suitable alliance and a firm agreement among seafood industries with future fish and shrimp silage producers are required to secure raw material (seafood waste) supply.

## 5. Conclusions

In Bangladesh, every year thousands of tons of fish and shrimp waste are produced in the seafood processing industries. Neglecting the utilization of seafood waste leads to severe forms of wastage. Currently, shrimp and fish wastes are sold for local consumption and feeding farm fish. Portions of air bladders, scales, fins, shrimp shells, and appendages are exported to some Asian countries. The availability of seafood waste to prepare silage and its demand from animal feed industries favor seafood waste-based silage production in Bangladesh. However, seafood-based silage production and utilization to produce animal feeds will be a new venture. Initially, Khulna and Chittagong could be the most suitable locations for pilot-based fish and shrimp silage production. The performance of seafood silage-based feeds for aquaculture and livestock species in Bangladesh needs to be studied further.

**Supplementary Materials:** The following are available online at https://www.mdpi.com/2071-1050/13/4/2372/s1, Supplementary files—1, 2, 3.

**Author Contributions:** Conceptualization, M.J.I. and O.R.P.; methodology, M.J.I.; formal analysis, M.J.I.; investigation, M.J.I., and O.R.P.; resources, M.J.I.; data curation, M.J.I.; writing—original draft preparation, M.J.I.; writing—review and editing, M.J.I. and O.R.P.; visualization, M.J.I.; funding acquisition, M.J.I. Both authors have read and agreed to the published version of the manuscript.

**Funding:** This research was funded by the Food and Agriculture Organization (FAO), grant number FMM/RAS/298/MUL, and the APC was funded by the University of Bremen, Germany.

**Institutional Review Board Statement:** Not applicable.

**Informed Consent Statement:** Not applicable.

**Data Availability Statement:** The data presented in this study are available on request from the corresponding author.

**Acknowledgments:** The authors acknowledge the Deputy Director, Fish Inspection and Quality Control, Ministry of Fisheries and Livestock, Bangladesh for his cordial help in information collection. The spokespeople of seafood processing companies and animal feed mills are greatly acknowledged for taking part in the face-to-face interview and survey process.

**Conflicts of Interest:** The authors declare no conflict of interest. The funders had no role in the design of the study; in the collection, analyses, or interpretation of data; in the writing of the manuscript; or in the decision to publish the results.

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
