# Peer review of "Seafood Waste Management Status in Bangladesh and Potential for Silage Production"

_sustainability, doi:10.3390/su13042372_

Round 1

Reviewer 1 Report

The authors presented an exciting study to estimate the amounts of seafood waste produced in Bangladesh and evaluate seafood waste-based silage production potentials. This manuscript is well written, and this study's aim meets the scope of Sustainability very well. Since this paper is considered a survey paper, there are no more technical defects regarding the experimental design. However, there are still several concerns that need to be addressed before acceptance.

  1. There are some typos in the text. For example, line 261, In 2017-168? Please double-check this manuscript.
  2. Although it is an exciting idea to convert fish waste to fish silage, the authors cannot convince me that this option shows economic advantages over the other options based on the facts listed in this manuscript.  
  3. I would suggest the authors show more studies that demonstrated that fish fed by fish silage is equal to/higher than the fish fed by fish meals.

Author Response

14 February 2021

Editorial Department

Sustainability

Subject: Submission of the revised manuscript.

Dear Prof. Dr. Marc A. Rosen and Ms. Dora Wang

We wish to thank you for the consideration of our manuscript and the valuable inputs from the reviewers. In the following, we provide an overview of our changes to the manuscript in response to the reviewers' input. Besides, we did substantial language editing with the help of a native English speaker to improve sentence structure, appropriate wordings, (content) fluency, and to remove grammatical errors throughout the manuscript. We believe we have been able to address all the reviewers' concerns. You can see all the changes we did in the "Manuscript File (Changes marked)" with track changes mode. We wish to thank the reviewers for their constructive reviews, which have greatly improved the manuscript's quality.

With best regards

Jakiul Islam

Response to Reviewer 1

The authors presented an exciting study to estimate the amounts of seafood waste produced in Bangladesh and evaluate seafood waste-based silage production potentials. This manuscript is well written, and this study's aim meets the scope of Sustainability very well. Since this paper is considered a survey paper, there are no more technical defects regarding the experimental design. However, there are still several concerns that need to be addressed before acceptance.

Authors’ response: Thank you for your kind comments and suggestions. We have emphasized all the comments raised by you and responded accordingly.  You can see our changes as the track changed marked in the “Manuscript File. Besides, we did substantial language editing with the help of a native English speaker to improve sentence structure, appropriate wordings, (content) fluency, and to remove grammatical errors throughout the manuscript. We believe we have been able to address all of your concerns.

  1. There are some typos in the text. For example, line 261, In 2017-168? Please double-check this manuscript.

Authors’ response: Our apologies for the typos. We did the correction (Line: 313). Besides, we have checked language throughout the manuscript to improve sentences and removing typos. You will see the changes in the manuscript. Thank you so much.

  1. Although it is an exciting idea to convert fish waste to fish silage, the authors cannot convince me that this option shows economic advantages over the other options based on the facts listed in this manuscript.

Authors’ response: Thank you so raise this point. We provided some information in the discussion section to support the cost-effectiveness of fish silage over other options (Lines: 342-367).

  1. I would suggest the authors show more studies that demonstrated that fish fed by fish silage is equal to/higher than the fish fed by fish meals.

Thank you for your nice suggestion. We have added text to justify fish silage is equal to/higher than the fish fed by fishmeal (Lines: 412-442). The text reads

“Research has shown that the nutritional benefits and economic feasibility of using fish silage as a good feedstuff for aquafeeds (Arvanitoyannis and Tserkezou, 2014). Inclusion of fish silage in feed increase the appetite and growth of terrestrial animals (Toppe et al., 2017). For example, a fish silage based diet fed to Nile tilapia, Oreochromis niloticus had good growth performance equivalent to a fishmeal diet (Wassef, 2005). Likewise, fish silage-based diets supplied to Nile tilapia O. niloticus (Madage et al., 2015) and African catfish C. garipinus evidenced 25 to 50% fishmeal replacement (Hanafy and Ibrahim, 2004). Feeding digestibility and growth studies have shown that fish silage is highly digestible and recorded effective replacement for up to 75% of fish meal in aqua-feeds (Vidotti et al., 2003). For shrimp, Litopenaeus vannamei fed with 25 to 50 % fish silage-based diets showed higher growth performance (Gallardo et al., 2012; Shao et al., 2020), gut health (Shao et al., 2019). Besides, fish silage not only acts as a feed ingredient, but also an essential feed additive (Olsen and Toppe, 2017). Thus for animal diets, fish silage can be used to replace 25 to 75% of fishmeal. Moreover, 100% fishmeal replacement is also possible if fish silage is used along with other protein-rich plant ingredients, for example, soybean meal (Gallardo et al., 2012), rice bran (Madage et al., 2015)”. Again, we wish to thank you to improve our manuscript greatly.

Reviewer 2 Report

There are several issues to be considered before publication.

I wonder how much they keep increasing the demand for frozen shrimp and fish products. "Since 2008, despite the global recession, international demand for frozen shrimp and fish products is high, and the number of seafood importing countries from Bangladesh is on the rise." This background needs to prove it based on the evidence (e.g., average increasing percent?) after the global recession.

Please define the waste management concept in the introduction.

Is information about the amount of seafood waste production and management status still lacking in Bangladesh only? Or all many countries' problem as well? The authors need to explain this sentence in detail. To provide the implications for waste management in the supply chain, I think lack of information sharing is the main problem in this industry. Not only in Bangladesh, but seafood industries in other countries also might have the same issues.

The authors need to explain why the supply chain is not well organized in fish silage production (Line 79).

When did you perform the survey and, did you consider pre-test before conducting the survey?

Table 1. The authors transformed the currency. Please specify the criteria date and year. 1.0 US$=80 BDT (based on when? and references).

This study needs to discuss the sustainability impact of fish and shrimp waste management in detail (Section 3.5. is an integral part of this paper, but the authors did not discuss sustainability issues in waste management). If there are existing literature review, please add the findings insight here as well for study contribution. In table 10, it seems only considers the environmental aspect. The authors want to discuss only this part, then please change the title name for section 3.5.

In the discussion section, it is hard to follow all contents. Please consider managerial implications and theoretical implications separately. Moreover, this industry is not well-organized supply chain framework, if the authors can show the efficient supply chain for this industry based on the key findings, it would be an excellent paper.

The conclusion section needs to have a summary of key findings.

Author Response

14 February 2021

Editorial Department

Sustainability

Subject: Submission of the revised manuscript.

Dear Prof. Dr. Marc A. Rosen and Ms. Dora Wang

We wish to thank you for the consideration of our manuscript and the valuable inputs from the reviewers. In the following, we provide an overview of our changes to the manuscript in response to the reviewers' input. Besides, we did substantial language editing with the help of a native English speaker to improve sentence structure, appropriate wordings, (content) fluency, and to remove grammatical errors throughout the manuscript. We believe we have been able to address all the reviewers' concerns. You can see all the changes we did in the "Manuscript File (Changes marked)" with track changes mode. We wish to thank the reviewers for their constructive reviews, which have greatly improved the manuscript's quality.

With best regards

Jakiul Islam

Response to Reviewer 2

There are several issues to be considered before publication.

Authors’ response: Thank you for the kind words and suggestions to improve our manuscript. We have agreed fully and followed every suggestion from you. We have emphasized all the comments raised by you and responded accordingly in the “Manuscript File” with track changes mode. Besides, we did substantial language editing with the help of a native English speaker to improve sentence structure, appropriate wordings, (content) fluency, and to remove grammatical errors throughout the manuscript. We believe we have been able to address all of your concerns.

I wonder how much they keep increasing the demand for frozen shrimp and fish products. "Since 2008, despite the global recession, international demand for frozen shrimp and fish products is high, and the number of seafood importing countries from Bangladesh is on the rise." This background needs to prove it based on the evidence (e.g., average increasing percent?) after the global recession.

Thank you to shed light on this issue. We intended to show an increased export trend since 2002 from Bangladesh. From 2008, the export amount was stable despite the recession. Table 11 justify this statement. However, we have made necessary corrections to make it clear (Lines: 47-51).

Please define the waste management concept in the introduction.

We have added the waste management concept in the introduction section (Lines: 56-63).

Is information about the amount of seafood waste production and management status still lacking in Bangladesh only? Or all many countries' problem as well? The authors need to explain this sentence in detail. To provide the implications for waste management in the supply chain, I think lack of information sharing is the main problem in this industry. Not only in Bangladesh, but seafood industries in other countries also might have the same issues.

We have made it clear (Lines 81-83). Besides, from our experience, we can say that if the waste generated by the seafood companies can bring revenues through sustainable management practices, processing companies will come forward to join waste management systems.

The authors need to explain why the supply chain is not well organized in fish silage production (Line 79).

In this line, we tried to explain the advantage of silage production over fishmeal (Lines: 89-92). For example, a fishmeal plan needs well-organized and constant supply channels to keep running the plants. However, for fish silage production this type of well-organized supply channel is not essential. As processing industries can process their waste by themselves in the processing premises or to a nearby area. We have discussed these in detail in the discussion section (Lines: 361-366).

When did you perform the survey and, did you consider pre-test before conducting the survey?

Thank you to spot this. We have added survey duration in the methodology section (Line: 125). We designed this research based on our gained experience.

Table 1. The authors transformed the currency. Please specify the criteria date and year. 1.0 US$=80 BDT (based on when? and references).

We have added time and reference (Line: 202). Thank you so much to improve our manuscript greatly.

This study needs to discuss the sustainability impact of fish and shrimp waste management in detail (Section 3.5. is an integral part of this paper, but the authors did not discuss sustainability issues in waste management). If there are existing literature review, please add the findings insight here as well for study contribution.

We have added text to discuss sustainability emphasizing the environment and economy (Lines: 274-285).

In table 10, it seems only considers the environmental aspect. The authors want to discuss only this part, then please change the title name for section 3.5.

Thank you to spot this error. We have removed economic impacts from section 3.5 title (Line: 271). Besides, we did the correction for Table 10, as we studied only environmental effects (Lines: 271-280).

In the discussion section, it is hard to follow all contents. Please consider managerial implications and theoretical implications separately.

We have tried to separate managerial and theoretical implications. However, to keep content fluency we have kept both implications together. We did the necessary edition (adding new text and removing some) to increase the content coherence (for example, Lines 341-366, 412-426, 432-443). You will our changes throughout the manuscript.

Moreover, this industry is not well-organized supply chain framework, if the authors can show the efficient supply chain for this industry based on the key findings, it would be an excellent paper.

Thank you for this nice suggestion. To avail this, based on our findings, we have proposed a supply chain flow chart (Figure 3, Line: 252; 235-240).

The conclusion section needs to have a summary of key findings.

We have revised the conclusion section to accommodate the summary of the key findings (Lines: 446-461). Finally, we wish to thank you for your kind comments, which have improved our manuscript greatly.

Round 2

Reviewer 2 Report

The updated version of the paper has addressed by comments.